# Structural basis of cooperativity in kinesin revealed by 3D reconstruction of a two-head-bound state on microtubules

**Daifei Liu, Xueqi Liu, Zhiguo Shang†‡, Charles V Sindelar\***

Department of Molecular Biophysics and Biochemistry, Yale University, New Haven, United States

**Abstract** The detailed basis of walking by dimeric molecules of kinesin along microtubules has remained unclear, partly because available structural methods have been unable to capture microtubule-bound intermediates of this process. Utilizing novel electron cryomicroscopy methods, we solved structures of microtubule-attached, dimeric kinesin bound to an ATP analog. We find that under these conditions, the kinesin dimer can attach to the microtubule with either one or two motor domains, and we present sub-nanometer resolution reconstructions of both states. The former structure reveals a novel kinesin conformation that revises the current understanding of how ATP binding is coupled to forward stepping of the motor. The latter structure indicates how tension between the two motor domains keeps their cycles out of phase in order to stimulate directional motility. The methods presented here pave the way for future structural studies of a variety of challenging macromolecules that bind to microtubules and other filaments.

**\*For correspondence:** charles.
sindelar@yale.edu

**Present address:** †Department of Cell Biology, University of Texas Southwestern Medical Center, Dallas, United States; ‡Department of Biophysics, University of Texas Southwestern Medical Center, Dallas, United States

**Competing interests:** The authors declare that no competing interests exist.

## Introduction

Kinesins are molecular motor proteins that use the energy of ATP hydrolysis to move unidirectionally along microtubules. Dozens of kinesin proteins have been found in humans, functioning in a wide variety of biological processes, from cellular cargo transport to mitosis (*Rath and Kozielski, 2012*). A commonly used nomenclature divides the kinesin family into 14 classes, based on phylogenetic analysis (*Lawrence et al., 2004*). In the current work, we focus on conventional kinesin-1. The kinesin-1 motor domain, which houses both ATPase and microtubule-binding activities, is located at its N-terminal end. Kinesin-1 forms a homodimer via the formation of a coiled coil by the stalk domains in the middle of the polypeptide chains. An important structural element, called neck linker, connects the motor domain to the coiled coil. During motility, kinesin-1 homodimer takes eight nanometer steps through head-over-head movements, advancing its two motor domains alternatively along a single protofilament (*Gennerich and Vale, 2009*).

To ensure alternate head stepping, the enzymatic cycles of the two motor domains must be kept out of phase. In particular, single-molecular optical trapping and bulk fluorescence analyses have suggested that, when both motor domains of a kinesin dimer are bound to a microtubule, biochemical processes at the leading head (binding and/or hydrolysis of ATP) are stalled until the trailing head detaches from the microtubule (*Andreasson et al., 2015*; *Clancy et al., 2011*). Such inter-head coordination is critical for effective motility, in part because premature ATP binding and/or hydrolysis in the leading head would be futile or could generate tension between the two heads and cause the kinesin dimer to fall off the microtubule. Strikingly different models to explain the inter-head coordination have been speculated based on structures of monomeric kinesins (*Morikawa et al., 2015*; *Shang et al., 2014*; *Wang et al., 2015*); however, due to a lack of detailed

structural information describing the microtubule-bound cycle of dimeric kinesin, the basis of the coordination has remained unclear.

Several attempts have been made to solve structures of dimeric kinesin on microtubules. However, in previous studies, the dimers were only resolved in a one-head-bound state with the partner head detached from the microtubules (*Hirose et al., 2000*; *Hoenger et al., 2000*). One possible cause of this observation is the over-saturating concentrations of kinesin dimers used (relative to microtubules). In addition, all previous studies relied on conventional helical reconstruction, and thus all kinesin-binding sites were averaged and any heterogeneity of kinesin dimer conformation (if present) was not properly accounted for (*Hirose et al., 2000*; *Hoenger et al., 2000*; *Egelman, 2015*).

To overcome these shortcomings, we developed a new method, FINDKIN, which is capable of locating bound kinesin motors on microtubules and has allowed us to determine the first structure of a kinesin dimer whose two motor domains bind at sequential sites along a single protofilament. This structure reveals asymmetric conformations of the leading and trailing heads that resemble the no-nucleotide and 'ATP-like' states of monomeric kinesin motor domains (*Cao et al., 2014*; *Gigant et al., 2013*; *Shang et al., 2014*) and suggests a mechanism of allosteric regulation of the motor enzymatic cycle by the neck linker. Unexpectedly, we also identified a second population of kinesin dimers in the same sample that attaches to the microtubule with only one motor domain. The structure determined from this latter population reveals that the attached motor domain assumes a novel conformation that is mostly ATP-like except that the neck linker is apparently disordered. These findings help reconcile a long-standing discrepancy between structural studies that seem to indicate that ATP-driven 'zippering' (or 'docking') of the neck linker represents a 'power stroke', and a variety of other biophysical measurements that fail to give evidence for this type of power stroke.

## Theory

For kinesin dimers in a two-head-bound state, the structural difference between the leading and trailing heads breaks helical symmetry and thus poses a major challenge to structural analysis. Conventional helical reconstruction, which has been the mainstay for solving monomeric kinesin structures (*Atherton et al., 2014*; *Morikawa et al., 2015*; *Shang et al., 2014*; *Sindelar and Downing, 2007*, *2010*), cannot be used because it treats all $\alpha\beta$-tubulin dimers as identical repeats and thus will average the associated leading and trailing heads together. Conventional single-particle analysis is also infeasible, due to the small size of kinesin motors and their densities being obscured by the microtubules. We developed a new method, FINDKIN, to address these problems.

FINDKIN is a combination of conventional helical and single-particle methodologies and can be divided into three major steps (*Figure 1A*). (i) Conventional helical analysis applied to microtubules yields helical parameters (XY-shifts and Euler angles) of all $\alpha\beta$-tubulin dimers in microtubules. (ii) FINDKIN determines which $\alpha\beta$-tubulin dimers are associated with kinesin motors. These tubulin dimers are termed occupied sites. (iii) Single-particle reconstruction using only the selected sites yields density map of kinesin dimers.

The key innovations of FINDKIN lie in the second step, where locations of kinesin motors on a microtubule are determined through a cross-correlation-based search algorithm (*Figure 1B,C*). To determine whether a particular pair of $\alpha\beta$-tubulin dimer in a microtubule is bound with kinesin or not, one strategy would be to compare the experimental image against two reference models differentiated by whether kinesin is present at this very site. However, multiple kinesin-binding sites frequently overlap in the experimental image, because it is a 2D projection of a 3D macromolecular structure, and these overlaps can potentially interfere with our analysis. For example, an imaged empty site may overlap with kinesin densities from neighboring occupied sites or even on the opposite side of the microtubule, causing simple correlation-based methods to misidentify the empty site as a kinesin-bound one and thus introducing a false-positive error. To overcome this difficulty, we expanded the search area used by FINDKIN to include all potential kinesin-binding sites that overlap with the $\alpha\beta$-tubulin dimer currently under examination (*Figure 1B*). Next, we enumerated all possible kinesin-binding patterns in the search area. As each $\alpha\beta$-tubulin dimer can have two states (occupied or empty), a search area of N $\alpha\beta$-tubulin dimers will have a total of $2^N$ binding patterns (*Figure 1C*). Cross-correlation scores were computed between the experimental image and $2^N$ reference images corresponding to $2^N$ binding patterns. The best-matched pattern is selected as the one that gives the highest score. A binding state (occupied or empty) was then assigned to the $\alpha\beta$-

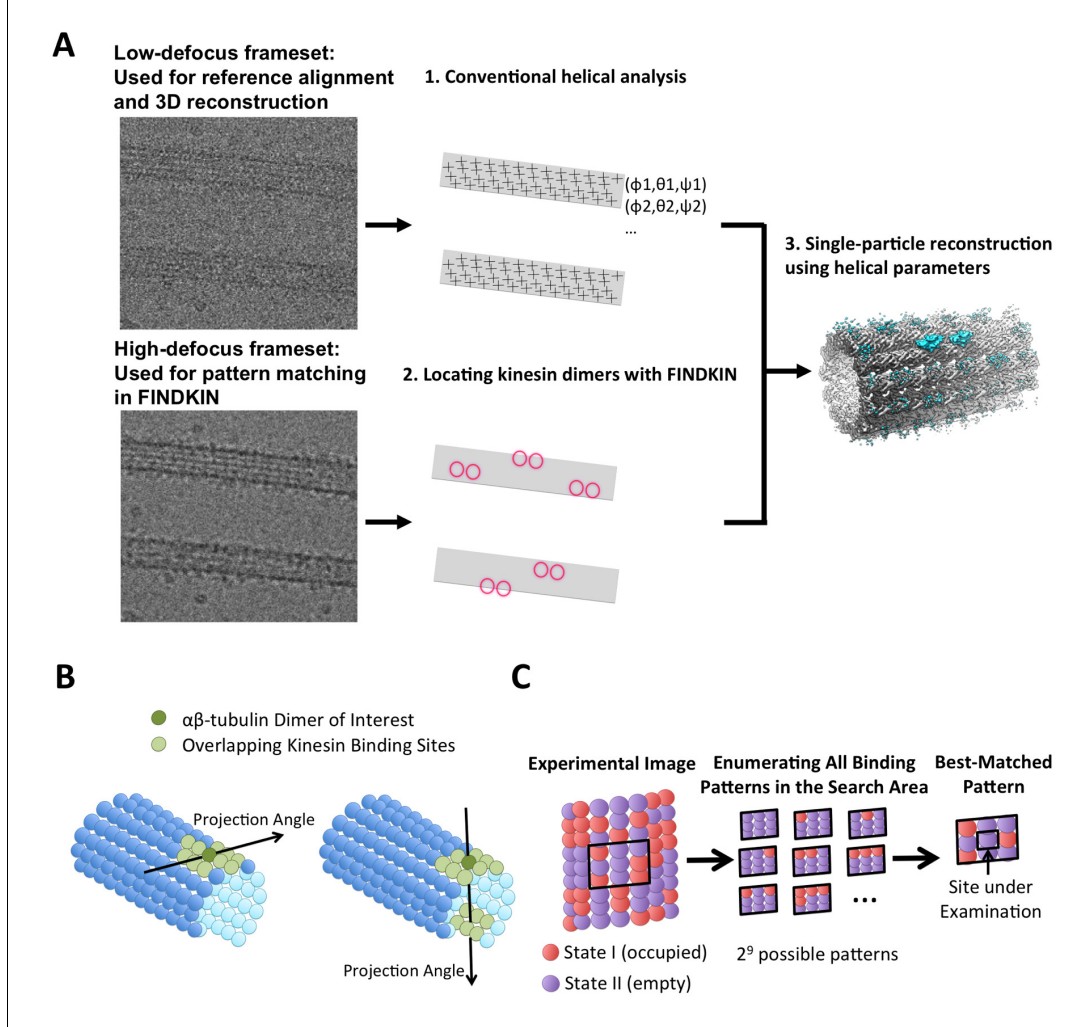

**Figure 1.** Schematic illustration of the FINDKIN algorithm. (**A**) Step I: Conventional helical analysis (IHRSR) is applied to low-defocus framesets to determine the XY-shifts and Euler angles of all tubulin dimers in the microtubule. Step II: FINDKIN determines which tubulin dimers are associated with kinesin with a cross-correlation-based search algorithm. Step III: Single-particle reconstruction using only the selected sites yields density map of dimeric kinesin. (**B**) Illustration of the search area associated with a particular pair of tubulin dimer under examination (shown in dark green). Other overlapping kinesin-binding sites in the search area are shown in light green. The search area is determined by the angle of projection. (**C**) Enumeration of all kinesin-binding patterns in the search area (shown as the black box). Two binding states (occupied or empty) are shown in red and purple. Some examples of the possible binding patterns in the search area are depicted. In this illustration, there are nine tubulin dimers in the search area, so there are $2^9$ possible binding patterns in total. The binding state of the site under examination is assigned according to the best-matched pattern. Raw images from our dataset and results from conventional helical analysis are shown in *Figure 1—figure supplement 1*. Specific details of the FINDKIN analysis are illustrated for a single microtubule in *Figure 1—figure supplement 2*. The FINDKIN algorithm is summarized in step-by-step flowcharts as shown in *Figure 1—figure supplement 3*.

The following figure supplements are available for figure 1:

**Figure supplement 1.** Raw Images and conventional helical reconstruction.

**Figure supplement 2.** Illustration of FINDKIN analysis for a single microtubule segment.

**Figure supplement 3.** Step-by-step flowcharts of the FINDKIN algorithm.

tubulin dimer under examination in accordance with the best-matched pattern (*Figure 1C*). The experimental image of one exemplary microtubule segment and the reference image for its best-matched pattern are illustrated in *Figure 1—figure supplement 2A–D*. This process was repeated for all αβ-tubulin dimers and the results were collectively recorded in kinesin-binding matrices, one for each microtubule. In the binding matrices, '1' stands for 'occupied site' and '0' stands for 'empty site'. Specific sites can then be selected for 3D reconstruction.

A potentially serious pitfall of the above strategy is that noise resembling kinesin densities may get selected, thus introducing noise bias by artificially causing the final 3D reconstruction to resemble the search model (*Henderson, 2013*). To circumvent this problem, we segregated each video taken with the K2 direct detector into two halves: a low-defocus frameset (first 10 frames) and a high-defocus frameset (remaining frames taken at a higher defocus). FINDKIN was applied to the high-defocus framesets in order to determine locations of occupied sites, which were then mapped back to the low-defocus framesets for 3D reconstruction (*Figure 1A*). Since noise originating from the camera is independent across video frames, this approach prevents model bias caused by camera noise from being carried into the final 3D reconstruction (note that while bias caused by sample noise, i.e. 'ice particles', etc., cannot be eliminated by this approach, camera noise is expected to predominate under typical low-dose cryo-EM conditions).

## Results

### FINDKIN is able to locate occupied sites on synthetic microtubules

In order to assess the efficacy of FINDKIN, we first applied it to synthetic microtubules, where the exact locations of bound kinesin motors are known and individual parameters, such as image noise level, occupancy (percentage of αβ-tubulin dimers bound with kinesin) and defocus can be controlled (*Figure 2—figure supplement 1*). By comparing the FINDKIN-generated kinesin-binding matrices and known locations of occupied sites, we were able to calculate false positive rates (number of false positive sites/total number of true sites) and false negative rates (number of missed sites/total number of true sites) and analyze how they are affected by the aforementioned parameters. At all noise, occupancy and defocus conditions tested, false negative rates remain below or around 0.2 (*Figure 2A,B*). By definition, false negative sites don't contribute to the final reconstruction and a 20% loss of occupied sites can readily be compensated by collecting more images. In contrast, the false positive rate goes up with increasing noise in the micrographs and goes down with higher occupancy (*Figure 2C,D*). Moreover the false positive rate is significantly affected by defocus. At low defocus (−1 to −2.5 μm), the false positive rate ranges from 0.4 for 40% occupancy to 1.2 for 10% occupancy at the highest noise level tested, while at high defocus (−7 to −8.5 μm), the rate stays at about 0.2, mostly insensitive to varying occupancy (*Figure 2C,D*). The lower false positive rate at high defocus is likely due to better preservation of low-resolution signal under these conditions by the contrast transfer function. Low-resolution signal contains more reliable information for the proper function of cross-correlation in FINDKIN, as high-resolution signal is often dominated by noise. This observation highlights the importance of utilizing high-defocus framesets for binding pattern matching in FINDKIN (*Figure 1A*).

### Application of FINDKIN to experimental dataset reveals dimeric kinesin on microtubules

We collected a dataset of dimeric kinesin (Kif5b aa 1–420) in complex with microtubules in the presence of 2 mM ADP•AlF$_x$. To prevent oversaturation of kinesin as seen in previous studies (*Hirose et al., 2000*; *Hoenger et al., 2000*), kinesin was mixed with microtubules at a ratio of ~0.6 motor domain per αβ-tubulin dimer. This led to substoichiometric binding, which was confirmed by the raw images (*Figure 1—figure supplement 1A,B*). Conventional helical reconstruction of microtubules (*Li et al., 2002*; *Shang et al., 2014*; *Sindelar and Downing, 2007*, *2010*) applied to our dataset yielded a high-quality map at ~5 Å resolution, with kinesin density much weaker than tubulins due to substoichiometric occupancy (*Figure 1—figure supplement 1C,D*). We only included 13-protofilament microtubules in subsequent FINDKIN analysis, because they are the most numerous in our dataset.

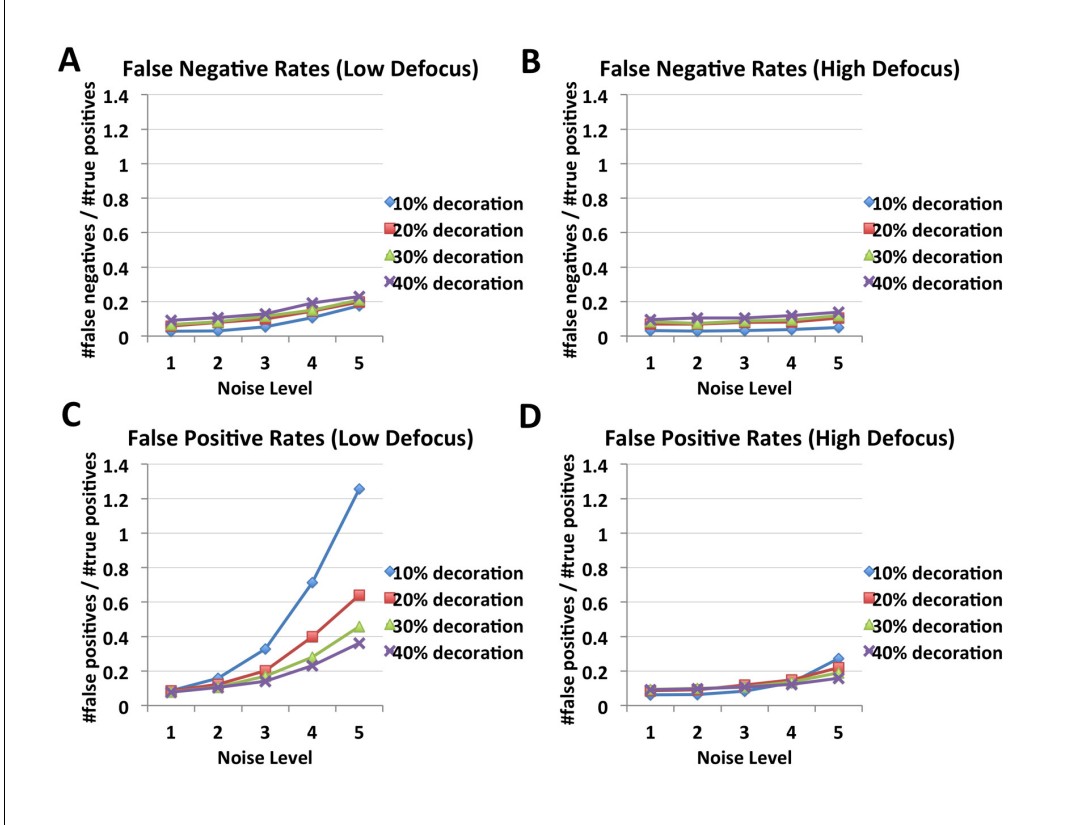

**Figure 2.** FINDKIN performances on synthetic microtubules, affected by noise level, occupancy and defocus. Panels (**A**) and (**B**) show the false negative rates of FINDKIN at low and high defocus respectively. False negative rates stay below or around 20%. Panel (**C**) shows false positive rates at low defocus, which increase dramatically with higher noise, especially at low occupancy. In contrast, panel (**D**) shows the false positive rates at high defocus, which reach around 20% even at the highest noise level tested. Examples of the synthetic microtubules are shown in *Figure 2—figure supplement 1*.

The following figure supplement is available for figure 2:

**Figure supplement 1.** Synthetic microtubules.

Next we applied FINDKIN in order to identify isolated pairs of kinesin motor domains that bind at consecutive sites along a single protofilament of the microtubule; this binding geometry corresponds to a '0110' pattern in the kinesin-binding matrices. However, preliminary 3D reconstructions using this approach revealed residual density corresponding to kinesin bound at the '0' sites (data not shown), which we attribute to the relatively high false negative rate of FINDKIN under the given experimental conditions. Due to this type of error, it is probable that FINDKIN misidentifies a relatively high proportion of alternative binding patterns (corresponding, i.e., to '1100', '0011', '1110', '0111', etc.) as matches for the target '0110' pattern. These misidentifications would therefore be expected to 'contaminate' the resulting 3D reconstruction of an isolated kinesin dimer with, for example, dimers that are shifted by ±8 nm along the protofilament (corresponding to '0011' and '1100' binding patterns, respectively; see *Figure 1—figure supplement 2F–H*). Such 'frame shift' errors would thus give rise to residual kinesin density at the '0' sites adjacent to the target dimer, explaining the above-mentioned artifacts in our preliminary 3D reconstructions.

In order to address this 'contamination' problem, we applied FINDKIN to the low-defocus framesets in addition to the high-defocus framesets, thus generating two independent binding matrices that could be combined in order to reduce error rates in our pattern identification (*Figure 1—figure supplement 2F–H*). Applying FINDKIN to the low-defocus framesets, however, introduces noise bias in the resulting 3D reconstruction (see theory section above); we therefore reduced the stringency of FINDKIN in the low-defocus framesets, so that the low-defocus binding matrices were only used to

enforce vacancies adjacent to the target dimeric heads ('0's in the '0110' pattern). Thus, in our final identification of '0110' sites from the binding matrices, kinesin motor domains are positively selected using only the high-defocus framesets while vacancies at the adjacent sites are selected using both the low- and high-defocus framesets. While this approach does introduce noise bias from the low-defocus framesets into the 3D reconstruction, it does so selectively at the vacant sites; noise bias within the protein region of our map is therefore minimized in our final method.

3D reconstruction of these '0110' sites yielded density corresponding to a pair of neighboring motor domains (*Figure 3A*; *Figure 3—figure supplement 1B*). Accordingly, we resolved secondary structural elements within the two heads that are consistent with the known architecture of the kinesin motor domain, including the switch II helix, central beta sheet and the switch loops (*Figure 4*). Our ability to resolve these structural elements indicates that the kinesin density in this map extends into the sub-nanometer range; secondary structure within the tubulin region of the map appears to be even better defined, indicating a slightly higher resolution in the microtubule than in kinesin. The reference model used in FINDKIN does not exhibit any secondary structure, because it is low-pass filtered to 20 Å resolution; thus, our ability to resolve secondary structure demonstrates that the kinesin features in our map do not arise from model bias (*Henderson, 2013*).

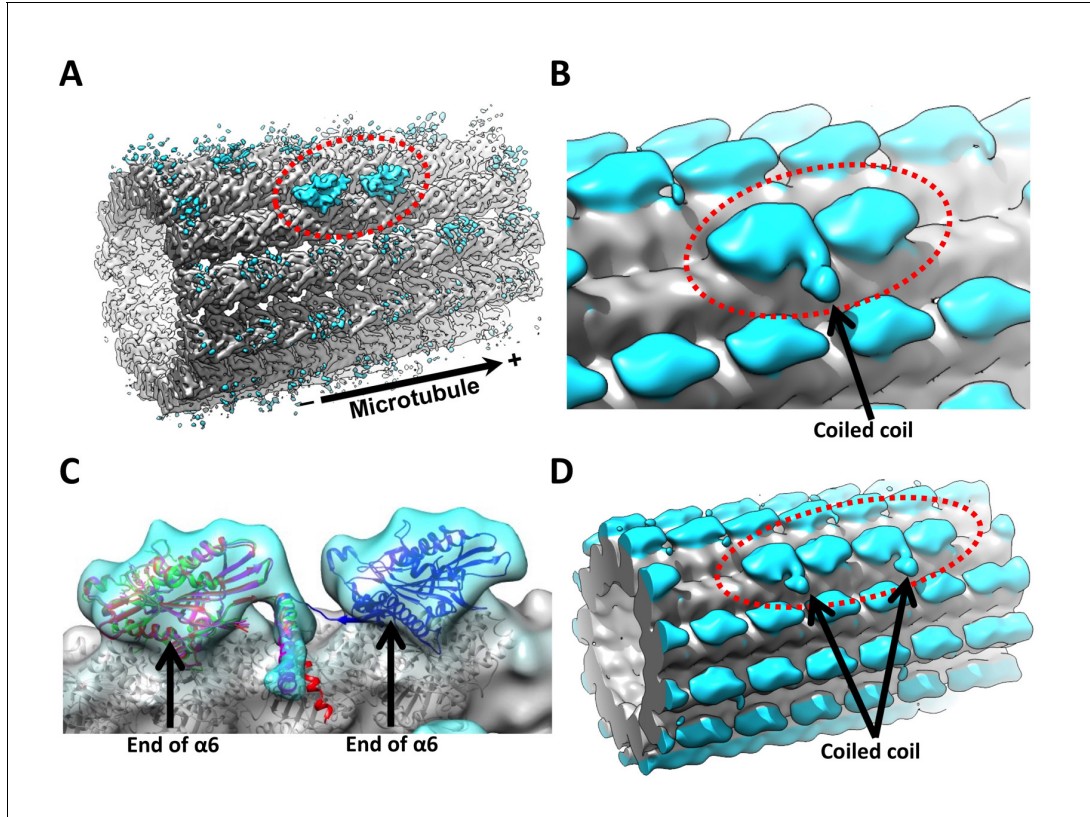

**Figure 3.** 3D reconstruction of kinesin dimer. (**A**) The kinesin dimer density map (low-pass filtered to 8.5 Å) shows two heads (circled with red dashed line) that bind sequentially along one protofilament. (**B**) Same map low-pass filtered to 15 Å. Arrow points to the density corresponding to the coiled coil. No contact between the coiled coil density and the microtubule surface is evident in our map. (**C**) Crystal structures of kinesin motor domain (PDB ID: 1MKJ, 3KIN) are fit into the trailing head. Direction of the coiled coil density in the map is consistent with the PDB models. Additionally, 3J8X is fit into the leading head. Arrows point to the ends of helices α6, where the neck linkers originate. The coiled coil density appears closer to the trailing head, because the end of helix α6 is closer to the minus-end side of the motor domain. The neck linker from 1MKJ can extend from the coiled coil to the ends of helices α6 in both heads. (**D**) Parallel 3D reconstruction derived from '011110' sites, showing two neighboring pairs of kinesin dimers. FSC curves are shown in *Figure 3—figure supplement 1*.

The following figure supplement is available for figure 3:

**Figure supplement 1.** FSC curves.

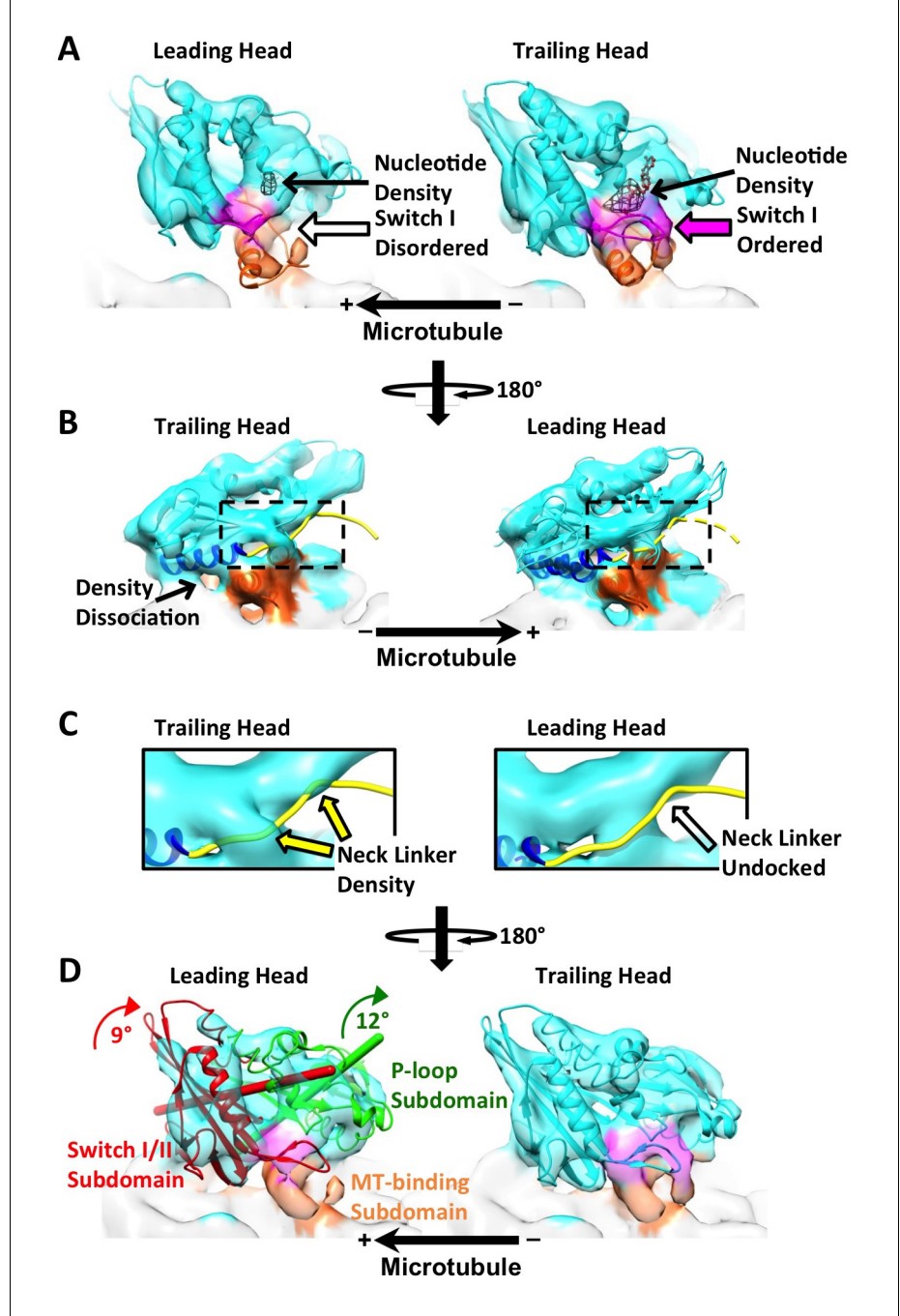

**Figure 4.** The leading and trailing heads show substantial conformational differences in the dimer map. Panels (**A**) and (**B**) are views from opposite angles. Density features corresponding to the switch I loop and the microtubule-binding subdomain are shown in magenta and orange respectively. In panel (**A**), two PDB models, 4LNU (no-nucleotide state) and 4HNA (ATP-like state), are fit into the leading and trailing heads respectively. Black mesh shows the nucleotide density. (Nucleotide corresponds to the strongest density feature in the kinesin map, so a higher threshold is chosen in Chimera to show only the nucleotide density). Solid and hollow magenta arrows denote ordered and absent density feature corresponding to switch I loop respectively. In panel (**B**), both 4LNU and 4HNA are fit into the leading head. Helix α6 is shown in blue. In the trailing head, this helix is tilted upwards, reflected by the dissociation of density with the microtubule. In the leading head, density is continuous between α6 and the microtubule surface. As a result, α6 from 4LNU and 4HNA can both be fit into the density so that its precise orientation is not clear from our map. To better visualize the conformational states of the neck linkers, regions within the dashed boxes are enlarged and shown in Panel (**C**). The neck linker from 4HNA is shown as a

*Figure 4 continued on next page*

*Figure 4 continued*

yellow strand. In the trailing head, portions of the density overlap with the neck linker in the PDB model (denoted by the solid yellow arrows), but not in the leading head. Panel (**D**) is in the same viewing angle as in Panel (**A**). In the leading head, ribbons of the P-loop and switch I/II subdomains are shown in green and red, with their respective rotation axis and magnitudes colored accordingly. Both subdomains are shown in cyan in the trailing head. Similar results are observed from a parallel reconstruction of kinesin dimer from the '011110' patterns, shown in *Figure 4—figure supplement 1*. Comparison of the subdomain rotations with published structures of monomeric kinesins is shown in *Figure 4—figure supplement 2*. Comparison of the kinesin conformations by cross-correlation is summarized in *Figure 4—figure supplement 3*.

The following figure supplements are available for figure 4:

**Figure supplement 1.** Parallel reconstruction of kinesin dimer derived from the '011110' sites.

**Figure supplement 2.** Kinesin subdomain rotations in the dimer map and in published monomeric structures.

**Figure supplement 3.** Cross-correlation scores between kinesin heads.

---

The FINDKIN algorithm, which uses monomeric kinesin for reference model, cannot distinguish between a kinesin dimer and e.g. two unassociated singly-bound heads that happen to lie next to each other. For a dimer, however, there will be a physical linkage between the two heads; moreover, previous studies have led to the general expectation that the two heads of a dimer will have different conformations. In contrast, two singly-bound heads should be unconnected and identical.

Two features in our map favor the former possibility. First, as will be discussed below, our map reveals that the leading and trailing heads assume distinct structural states, which further corroborates our identification of a kinesin dimer in the map. Second, an extended density feature was resolved between the two heads, corresponding to the coiled coil dimerization domain (*Figure 3B*). Several crystal structures of the kinesin motor domain include portions of the neck region that forms the coiled coil (PDB ID: 1MKJ, 3KIN), and the orientation of the neck in these crystal structures is generally consistent with the coiled coil density in our dimer map (*Figure 3C*) (*Kozielski et al., 1997*; *Sindelar et al., 2002*). The coiled coil density appears to locate equidistantly from the C-terminal ends of helices α6 (where the neck linkers originate) in both heads, indicating that the neck linkers in the leading and trailing heads extend for about the same length (*Figure 3C*) (note that because the C-terminal end of α6 is closer to the minus-end side of the motor domain, the coiled coil density thus appears closer to the trailing head). The coiled coil density does not appear to contact the microtubule surface in our map. While the coiled coil density is evident after filtering the map to 15 Å resolution, this feature is not apparent at sub-nanometer resolution, suggesting greater flexibility of this region compared to the bound motor domains. Consistent with this observation, the crystal structures show substantial variability in the neck orientations (*Figure 3C*). Thus, the behavior of the coiled coil in our specimen appears to be consistent with expectations based on previous structural studies (note that, despite the above analysis, we cannot rule out that our dimer map may be contaminated by two adjacently bound singly-bound motor domains to some extent; see discussion section for details).

Based on the above behavior for the isolated dimer ('0110' pattern), it follows that searching for four consecutively bound motor domains in our images ('011110' pattern in the kinesin-binding matrices) should reveal the structure of two consecutive kinesin dimers along one protofilament. Consistent with this prediction, density for the coiled coil is resolved between the first and second heads and also between the third and fourth heads, indicating that two kinesin dimers are bound next to each other (*Figure 3D*). We were able to reconstruct independent 3D maps of the kinesin dimer from both '0110' and '011110' patterns, providing independent validation for our structural findings as discussed below.

## Leading and trailing heads exhibit distinct chemical and conformational states

In order to test whether the two motor domains of a kinesin dimer assume different conformations, we performed cross-correlation analysis between leading and trailing heads of our two independent dimer maps (*Figure 4—figure supplement 3B*). The cross-correlation scores between the two leading heads and between the two trailing heads (0.91 and 0.92, respectively) are consistently higher than the scores between a leading and a trailing head, either from the same map or between two maps (range from 0.86 to 0.88). These comparisons thus confirm that the leading and trailing heads of the kinesin dimer are structurally distinct from each other, and that our 3D reconstruction methods are capable of resolving these differences.

Closer examination of the dimer maps reveals that the leading head exhibits substantially weaker density in the nucleotide cleft, when compared with the trailing head, thus indicating preferential binding of nucleotide to the trailing head (*Figure 4A*; *Figure 4—figure supplement 1A*). Reduction of nucleotide density in the leading head is accompanied by a near-total loss of density corresponding to the switch I loop, which directly participates in coordination of the γ-phosphate as well as hydrolysis (*Kull et al., 1996*; *Sablin et al., 1996*). In contrast, density for the switch I loop in the trailing head is well resolved, consistent with a closed nucleotide cleft as observed in previous structural studies with ATP analogs (*Figure 4A*; *Figure 4—figure supplement 1A*) (*Gigant et al., 2013*; *Shang et al., 2014*; *Hirose et al., 2006*; *Kikkawa and Hirokawa, 2006*; *Sindelar and Downing, 2010*). Thus, these features demonstrate that symmetry between the two heads is broken at the nucleotide cleft.

It is generally expected that the neck linker in the trailing head docks along the motor domain towards the leading head, while the neck linker in the leading head extends back towards the trailing head in a disordered conformation (*Rice et al., 1999*). Multiple features in our maps support this prediction. First, we directly resolved weak density corresponding to the docked conformation of the neck linker in the trailing head in our map. In contrast, neck linker density is not evident in the leading head, indicating that its neck linker is structurally flexible (*Figure 4B,C*; *Figure 4—figure supplement 1B,C*). Second, the C-terminal end of helix α6, where the neck linker is attached, is shifted upwards in the trailing head, reflected in our maps by the loss of contact between this helix and the microtubule surface (*Figure 4B*; *Figure 4—figure supplement 1B,C*). In contrast, the leading head displays continuous density in this region, thus indicating that α6 approaches the microtubule surface more closely (*Figure 4B,C*; *Figure 4—figure supplement 1B,C*). A similar correlation between the position of α6 and undocking of the neck linker is observed in co-complexes of monomeric kinesin and αβ-tubulins (*Cao et al., 2014*; *Gigant et al., 2013*; *Shang et al., 2014*; *Kikkawa and Hirokawa, 2006*; *Sindelar and Downing, 2010*), strongly supporting our identification of a docked neck linker in the trailing head and an undocked one in the leading head.

Recent structural studies of monomeric kinesin bound to tubulins or microtubules have demonstrated that neck linker docking is associated with concerted rearrangements of three distinct subdomains of kinesin (*Cao et al., 2014*; *Shang et al., 2014*) (*Figure 4D*). In particular, neck linker docking is coupled with a seesaw-like rotation of the P-loop subdomain to which the neck linker is attached. In order to test whether there is similar motion in the leading and trailing heads in our dimer maps, we fit a truncated PDB model corresponding to the P-loop subdomain (PDB ID: 3J8Y aa 9–66, 76–115, 293–321) into the kinesin head densities and measured the rotation relative to a previously published EM map of monomeric kinesin in an ATP-like state (EMDB ID: 6188) (*Shang et al., 2014*). The measured rotation of the P-loop subdomain in the leading head was 12–15°, while no significant rotation was found in the trailing head (*Figure 4D*; *Figure 4—figure supplement 1D*). A smaller rotation (5–9°) was likewise identified in the switch I/II subdomain of the leading, but not the trailing head. Although the magnitude of these subdomain rotations in the leading head are smaller than was measured in prior studies with monomeric kinesin in the no-nucleotide state, the orientation of the rotation axes are approximately consistent (*Figure 4—figure supplement 2B,F*) (*Cao et al., 2014*; *Shang et al., 2014*). Functionally these rotations will close the docking cleft and sterically obstruct neck linker docking in the leading head.

The above observations are consistent with an 'alternating cleft' model, proposed based on prior studies of monomeric kinesins, which posits that the trailing head of a kinesin dimer will have a closed nucleotide cleft and an open docking cleft, while the leading head will have the opposite (an

open nucleotide cleft with reduced affinity for nucleotides and a closed docking cleft that precludes neck linker docking) (*Shang et al., 2014*; *Wang et al., 2015*). The current work provides direct support for this model by visualizing the conformations of the clefts in dimeric kinesin.

The above features of the trailing head, including a closed nucleotide cleft and a docked neck linker, indicate that this head may be well modeled by structures of monomeric kinesins co-complexed with ATP analogs (*Gigant et al., 2013*; *Shang et al., 2014*). On the other hand, the leading head shares certain features with structures of monomeric kinesins in the no-nucleotide state, including an open nucleotide cleft with disordered switch I loop and a closed docking cleft (*Cao et al., 2014*; *Shang et al., 2014*). We therefore assessed the overall resemblance between our kinesin dimer map and available cryo-EM maps of monomeric kinesins (EMDB: 6187, 6188 for no-nucleotide and ATP-like states, respectively; all maps low-pass filtered to 9 Å). We aligned the tubulin densities in the maps of monomeric kinesins and our dimer maps and computed cross-correlation scores between the kinesin densities (summarized in graph form in *Figure 4—figure supplement 3A*). In line with the above analysis, the trailing heads resemble the ATP-like state of monomeric kinesin (EMDB: 6187) more closely than the no-nucleotide state (EMDB: 6188) (cross-correlation scores were 0.73–0.74 and 0.67, respectively), and the leading heads resemble the no-nucleotide state of monomeric kinesin more closely than the ATP-like state (cross-correlation scores: 0.77–0.78 versus 0.69, respectively). These correlation scores are consistent with the 'alternating cleft' model.

## Identification of singly-bound heads in a novel conformation

Preceding work has not definitively established whether one or both heads of a kinesin dimer will attach to microtubule in the presence of ADP•AlF$_x$. In order to address this question, we searched for isolated motor domains with no neighboring head along the same protofilament ('010' pattern in the kinesin-binding matrices). Reconstruction of these sites identified kinesin dimers that attach to microtubules with only one head, which are termed singly-bound heads (*Figure 5*). While in principle these '010' sites could reflect non-canonical stepping configurations, in which the partner head has stepped diagonally onto an adjacent protofilament, the absence of enriched kinesin density at diagonally adjacent sites indicates that this possibility is unlikely in our sample (*Figure 5A*).

Because detachment of the partner head removes stereochemical constraints, under our experimental conditions, a singly-bound head might be expected to assume an ATP-like state as previously observed in monomeric kinesin with ADP•AlF$_x$, including nucleotide binding, an ordered switch I loop and a docked neck linker (*Gigant et al., 2013*; *Shang et al., 2014*). We observed the first two features in the singly-bound head (*Figure 5B*); moreover, when we quantified the orientations of the switch I/II and P-loop subdomains, they were indistinguishable from those seen in the trailing head of our dimer maps (*Figure 4D*; *Figure 4—figure supplement 2C*). However, density for the neck linker in the singly-bound head is not evident and continuous density is observed between helix α6 and the microtubule surface, similar to the leading head (*Figure 5C*). Moreover, there is no trace of density for the coiled coil in this map (*Figure 5A*). Thus, the singly-bound head appears to represent a novel conformation of kinesin that is mainly ATP-like but with a disordered neck linker.

## Discussion

### Role of the neck linker in inter-head cooperativity

As we have shown, the trailing and leading heads in our kinesin dimer map closely resemble previously determined structures of monomeric kinesins in ATP-like and no-nucleotide states, respectively. However, it is important to note that the underlying mechanistic causes of the conformational differences between the two heads of a dimer in the current work and between the two nucleotide states of monomeric kinesin are not the same. In previous studies with monomeric kinesin, the motor domains were locked into a certain conformation by the nucleotide conditions used in the experiments. More specifically, AlF$_x$ was used to lock the motor domain in an ATP-like state (ADP•AlF$_x$), or apyrase was used to remove all nucleotides to ensure a no-nucleotide state (*Cao et al., 2014*; *Gigant et al., 2013*; *Shang et al., 2014*). In contrast, the two heads in our dimer sample are biochemically identical and are exposed to the same level of ATP and AlF$_x$ in the environment, and yet strikingly they exhibit different conformations and different levels of nucleotide binding. The only

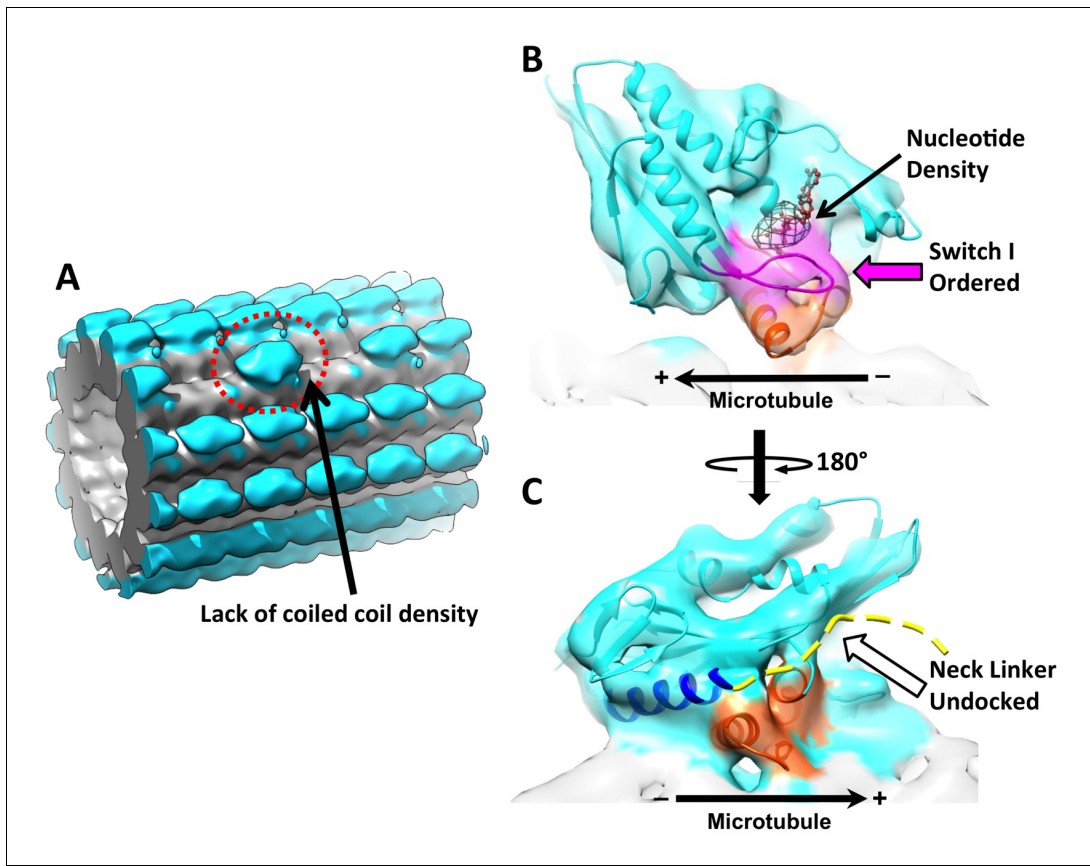

**Figure 5.** 3D reconstruction of '010' sites reveals singly-bound heads. (**A**) The singly-bound head is circled with red dashed line. Panels (**B**) and (**C**) follow the same labeling and formatting as in *Figure 4*. (**B**) The singly bound head adopts a closed conformation of the nucleotide cleft. (**C**) There is continuous density between helix α6 and the microtubule surface and no density overlaps with the neck linker strand in 4HNA fit into the map.

logical explanation for this observation lies in the different orientations of the neck linkers that connect the two heads, since all other factors appear identical in our experimental conditions.

The alternating cleft model provides a structural explanation for understanding this behavior. Due to inter-head geometric constraints, the neck linker in the leading head must orient rearward to connect with the trailing head. As a result of this rearward orientation, neck linker docking cannot be achieved in the leading head and thus a closed docking cleft will be favored, leading to a seesaw-like rotation of the P-loop subdomain that opens up the nucleotide cleft concomitantly. We have now directly visualized evidence of this coupling between the docking cleft and the nucleotide cleft due to rearwards pulling of the neck linker. Our results therefore tend to disfavor a competing 'water cap' model (*Morikawa et al., 2015*; *Nitta et al., 2008*) that involves melting of the C-terminus of the switch II helix and other structural features that are not observed in the current dimer map.

### Contamination with singly-bound heads may largely explain discrepancies between the kinesin dimer map and structures of monomeric kinesins

We observed several significant discrepancies between the kinesin dimer map and previously determined structures of monomeric kinesins that underlie the alternating cleft model (*Shang et al., 2014*). First, density corresponding to the nucleotide can still be observed in the leading head (albeit weaker compared to the trailing head), contrasting with the general expectation that nucleotide binding in the leading head should be strongly disfavored in the two-head-bound state. Second, the

magnitude of rotation of the P-loop subdomain is not as large in the leading head as in monomeric kinesin in the no-nucleotide state, although the orientations of the rotation axes are approximately consistent. These discrepancies could indicate that ATP still binds in the leading head to some extent, albeit with reduced affinity, in which case our leading head density will reflect a mixture of ATP-bound and -unbound states. To the best of our knowledge, no published work can solidly rule out this possibility (*Asenjo et al., 2003*; *Isojima et al., 2016*; *Mickolajczyk et al., 2015*).

An alternative explanation for the above discrepancies is that they arise from a contamination artifact, i.e. due to incomplete sorting of kinesin dimers by FINDKIN. Due to its design, the FINDKIN algorithm is highly effective at aligning and reconstructing kinesin dimers if they all exist in a two-head-bound state. However, the presence of singly-bound heads in our dataset (where the partner head is detached from the microtubules) introduces the possibility that two singly-bound heads may occupy adjacent sites along one protofilament, a configuration FINDKIN will falsely identify as a doubly-attached dimer. We therefore expect that densities of both leading and trailing heads in our map will be 'contaminated' through mixing with the singly-bound heads.

In order to test whether features in the leading head density may be explained by such a contamination artifact, we generated a model of a 'contaminated' leading head by averaging the monomeric kinesin density map in the no-nucleotide state and the singly-bound head, and this mixture model displays increased nucleotide density and a smaller rotation of the P-loop subdomain (*Figure 4—figure supplement 2H*). Moreover, the trailing head also differs from monomeric kinesin in the ATP-like state in that its neck linker density is weaker. This latter effect may also be explained by contamination with the singly-bound head, which exhibits no neck linker density. The presence of such 'contamination' could also help explain why cross-correlation scores that compare motor domains from our dimer maps with each other (ranging from 0.86 to 0.92; see *Figure 4—figure supplement 3B*) are substantially higher than scores that compare dimeric motor domains with monomeric ones (ranging from 0.67 to 0.78; see *Figure 4—figure supplement 3A*): by causing trailing and leading heads to be averaged with a common structure (the singly-bound head), the contamination artifact would tend to artificially increase the similarity of the leading and trailing heads in our dimer maps.

It therefore appears likely that at least some of the discrepancies between our dimer map and the previously determined structures of monomeric kinesins can be attributed to 'contamination' with the singly-bound heads. The other proposed explanation (ATP binding in the lead head), however, is not mutually exclusive with this contamination artifact, and there appears to be no clear indication from published work that either one is more probable. Further experiments are required to characterize key properties of the leading head, including the kinetics/affinity of ATP binding as well as the effects of ATP binding on the conformation of the nucleotide cleft.

## Features in the singly-bound head indicate that neck linker docking is destabilized when the partner head is unbound

Singly-bound heads of kinesin dimers have previously been inaccessible to high-resolution structural techniques, but are readily identified by FINDKIN in our dataset. The presence of singly-bound heads was not originally anticipated in our experimental conditions, as other studies of dimeric kinesin with various nucleotide analogs have generally been interpreted to show mainly two-head binding (*Kawaguchi and Ishiwata, 2001*; *Kawaguchi et al., 2003*; *Asenjo et al., 2003*; *Guydosh and Block, 2006*). However, a recent study from Mickolajczyk *et al.* showed that, with ATPγS, the kinesin dimer persists in a one-head-bound state for most of the motility cycle, indicating that the relative proportion of singly- versus doubly-attached states of kinesin dimers may vary depending on the kind of nucleotide analog used in the study (*Mickolajczyk et al., 2015*). Currently, it is impossible to directly correlate our experimental conditions with the aforementioned biophysical experiments, because evaporation following the final blotting step before plunge freezing of the grid can increase the concentrations of nucleotide analogs and salts in our sample to a nearly arbitrary extent, thus rendering it impossible to control for ionic strength, etc. The fact that we were able to identify both two-head-bound and singly-bound dimers in the current dataset indicates that there may be an equilibrium between the two states in our experimental conditions (*Figure 6A*). Further experiments are required to resolve this question and to determine the significance of such behavior with respect to the motility cycle of kinesin.

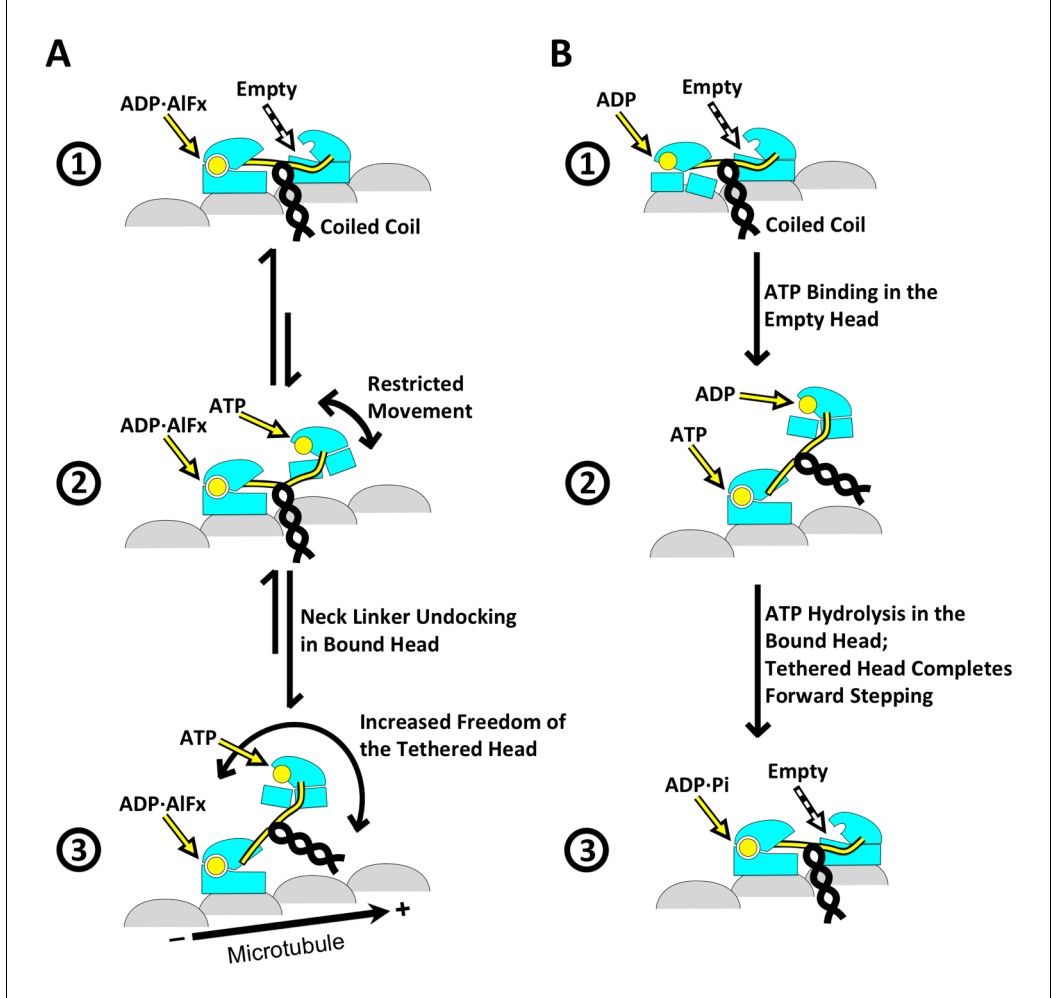

**Figure 6.** Schematic of conformational and chemical states of dimeric kinesins. (**A**) Model for the conformational and chemical states of dimeric kinesins with ADP•AlF$_x$ in our sample. State 1 corresponds to kinesin dimer in the two-head-bound state. State 2 corresponds to a highly transient state where the leading head detaches from the microtubule but the neck linker of the trailing head remains docked. This state transitions into state 3, driven by the entropy gain from a bigger space that the tethered head can explore once neck linker of the trailing head is undocked. State 3 corresponds to the singly-bound head. (**B**) Model for three key states in the regular motility cycle of dimeric kinesin with ATP. In state 1, after Pi releases, the trailing head (bound with ADP) has low affinity for the microtubule. After ATP binding in the leading head, the kinesin dimer enters a transitional state 2, where the nucleotide cleft is closed in the microtubule-bound head, but its neck linker is undocked. This state can be modeled by the singly-bound head. After ATP hydrolysis, the tethered head completes forward stepping, as seen in state 3. This state can be modeled by our two-head-bound kinesin dimer.

As discussed in the results section, the singly-bound head appears to represent a novel conformation that is largely ATP-like but with a disordered neck linker. The undocking of the neck linker may be entropy-driven. It is known that the thermodynamics of neck linker docking involves balance of strong opposing forces, including hydrophobic interactions between a docked neck linker and the opened docking cleft as well as high entropy gain from a disordered neck linker (*Rice et al., 2003*). In previous structures of monomeric kinesin with ADP•AlF$_x$, the neck linker is predominantly docked, but in the case of a singly-bound head, a free partner head may increase the entropy gain from a disordered neck linker, favoring an undocked state (*Figure 6A*). Another explanation for the disordered neck linker in the singly-bound head may be that neck linker docking with ADP•AlF$_x$ is intrinsically weak and the strong neck linker density observed in structures of monomeric kinesin is an artifact

due to strong interaction between the truncated coiled coil and the microtubule surface (*Shang et al., 2014*). In either case, the observed conformation of the singly-bound head serves as further evidence against the proposal that the neck linker delivers a power stroke that contributes significantly to force generation (*Vale and Milligan, 2000*). Rather, our results indicate that nucleotide-triggered docking of the neck linker for the kinesin dimer may be even weaker than was measured in EPR studies of monomeric kinesin (*Rice et al., 2003*), reinforcing the argument that nucleotide-dependent docking of the neck linker provides only a weak bias for directional movement. These observations are consistent with the recent study from Isojima *et al*., using high-speed imaging of nanogold-labeled kinesin dimers, that ATP binding is associated with little or no movement of the tethered head (*Isojima et al., 2016*).

In addition, the structure of the singly-bound head implies that full neck linker docking may not be required for hydrolysis, because the nucleotide cleft can be fully closed even when the neck linker is at least partially disordered. This observation indicates that kinesin may readily hydrolyze ATP prior to forward stepping by the tethered head. In accordance with this idea, several recent kinetic studies have shown that the tethered head of a kinesin dimer may not complete forward stepping until after hydrolysis has occurred (*Mickolajczyk et al., 2015*; *Milic et al., 2014*). Further structural work of kinesin in the post-hydrolysis state (e.g. ADP•Pi) will be needed to understand this key intermediate step in the motility cycle.

## Implications for the stepping mechanism of kinesin dimer

Two structural states of kinesin dimers were identified with ADP•AlF$_x$ in our sample (two-head-bound and singly-bound) that we propose can serve as models for two consecutive states in a regular motility cycle with ATP (*Figure 6B*; *Video 1*). As depicted in the model figure, ATP binding in the microtubule-bound head would favor closure of the nucleotide cleft, but not force docking of the neck linker. We propose that this state, which would be highly transient in the regular cycle, may be modeled by our map of the singly-bound head. Subsequent docking of the neck linker, potentially accelerated by ATP hydrolysis (*Milic et al., 2014*; *Mickolajczyk et al., 2015*), would create a kinetic pathway that allows forward stepping of the tethered head. Following binding of the tethered head to the microtubule, the rearward orientation of its neck linker would favor opening of the nucleotide cleft, leading to ADP release. As modeled by our two-head-bound dimer map, ATP binding/hydrolysis in the leading head would then be gated until the trailing head detaches from the microtubule. Future structural studies of dimeric kinesin with ATP will be needed to directly visualize the sequence of states in the processive motility cycle.

## Prospects for the study of heterogeneous filaments

Biological filaments are very often heterogeneous in nature. Even repeating subunits in a single filament may have different conformations or are bound with different proteins. These violations of helical symmetry pose great challenges to structural analyses. Conventional methods for helical reconstruction, with Iterative Helical Real Space Reconstruction (IHRSR) currently being the most widely used, do employ a single-

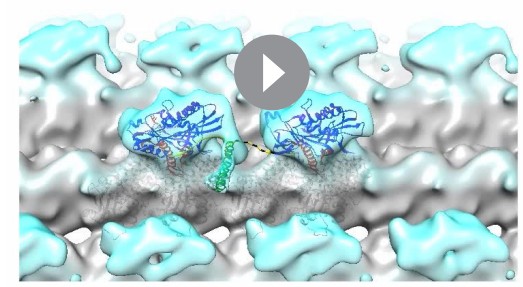

**Video 1.** Animation depicting our current model of the kinesin motility cycle, based on cryo-EM maps of dimeric kinesins in both two- and one-head bound states, in combination with previous structural and biochemical studies. This video highlights two principle features of this model. First, in the two-head-bound state, inter-head tension keeps the neck linkers in the two heads pointing in opposite directions; in the leading head, backward orientation of the neck linker favors an open nucleotide cleft, thus discouraging ATP binding/hydrolysis. Second, ATP binding in the one-head-bound state closes the nucleotide cleft, but fails to dock the neck linker (precise structural details of this novel conformational state remain unclear from the current data). Consequently, ATP hydrolysis is required to commit to a forward step, returning to the beginning of the depicted motility cycle.

particle approach, but they treat an artificially boxed segment of the filament as one particle (*Egelman, 2015*). As each segment contains many repeating subunits, heterogeneity within each segment cannot be accounted for. The present work demonstrates that it is possible to sort individual subunits of filaments into distinct structural classes, and thus FINDKIN represents an important advance over current methods of studying helical polymers.

More work is needed to broaden the application of FINDKIN. Currently, FINDKIN requires prior information of the different conformational states of the repeating subunits to generate reference models for comparison with the experimental images. Thus applications of FINDKIN are limited to situations where reasonably accurate guesses of the subunit structures are available. Future work will incorporate innovations of FINDKIN into likelihood or Bayesian frameworks in order to improve the ability to discriminate between different structural states and reduce the dependency on initial models.

## Materials and methods

### Kif5b aa 1–420 expression and purification

The plasmid for Cys-lite, dimeric human Kif5b aa 1–420 (from Sarah Rice, Northwestern University Feinberg School of Medicine) was transformed into BL21 (DE3) competent cells. Cells were allowed to grow to O.D.=1 before culture temperature was lowered to 24°C and IPTG was added to a final concentration of 200 µM. Cells were harvested after continuous incubation overnight. The cell pellets were resuspended in 50 mL Buffer A, containing 25 mM HEPES (pH 6.8), 2 mM $MgCl_2$, 1 mM EGTA, 1 mM TCEP, 0.02% Tween-20 and 20 µM ATP, supplemented with 1 tablet of cOmplete Protease Inhibitor Cocktail (Roche, Basel, Switzerland). French press was used to lyse the cells. The clear lysate obtained after centrifugation was loaded on a phosphocellulose column. The column was washed with Buffer A containing 100 mM NaCl and eluted with a gradient of increasing NaCl concentration of up to 1 M. Peak fractions containing K420 were pooled and diluted with Buffer A to lower NaCl concentration to 150 mM. Then the protein was loaded on a HiTrap SP HP column (GE Healthcare, Chicago, Illinois) and flow-through was collected (K420 doesn't bind in this condition). The flow-through was then diluted with Buffer A to lower NaCl concentration to 80 mM and loaded on a HiTrap Q HP column (GE Healthcare). The column was washed with Buffer A and eluted with a gradient of increasing NaCl concentration of up to 400 mM. Peak fractions containing K420 were pooled. Sucrose was added to the purified K420 at 20% before snap freezing in liquid nitrogen. K420 can be stored long-term at −80°C.

### Microtubule preparation

Microtubules batches were grown as previously reported (*Shang et al., 2014*). 250 µg of lyophilized bovine tubulin (Cytoskeleton, Denver, Colorado) was resuspended in 25 µL EM buffer containing 25 mM PIPES (pH 6.8), 25 mM NaCl, 1 mM EGTA and 2 mM $MgCl_2$, supplemented with 1 mM GTP. Dissolved tubulin was clarified (100 K RPM, 4°C, 10 min) followed by incubation at 37°C. After 10 min, equimolar level of taxol was added. Incubation was continued at 37°C for 45 min. Then, the microtubule solution was pelleted through a glycerol cushion (50 µl EM buffer +60% glycerol wt/vol +200 µM taxol) to remove unpolymerized tubulins (50 K RPM, 24°C, 20 min). The microtubule pellet was resuspended in 10 µl EM buffer plus 200 µM taxol.

### Cryo-EM sample preparation

K420 aliquot was thawed, exchanged into EM buffer through gel filtration with a Superose 6 10/300 GL column (GE Healthcare) and concentrated with Microcon ultracentrifugal filter (Merck Millipore, Billerica, Massachusetts). Concentrations of K420 and microtubules are determined by UV absorption. 37 µg of microtubules and 12.5 µg of K420 were mixed and diluted two-fold with water. The microtubule-K420 mix was then pelleted (13 K RPM, 24°C, 8 min) and resuspended in 5 µl two-fold diluted EM buffer. 0.7 µl of the resuspended mix was supplemented with 0.4 µl of 20 mM ATP, 0.4 µl of 20 mM AlCl3, 0.4 µl of 100 mM NaF and 2.1 µl of 4 mM MgCl2 to form a 4 µl-grid droplet and applied to Quantifoil holey carbon grids with 1.2 µm hole diameter and 1.3 µm spacing (Quantifoil Micro Tools GmbH, Großlöbichau, Germany) without glow discharge. The sample was allowed to incubate on the grid for 1 min and then most of the buffer was wicked away by touching the grid

edgewise with a piece of filter paper. The grid was then mounted on a home-built plunge-freezing apparatus, blotted completely and plunged into liquid ethane.

## Data collection

All micrographs were collected on FEI Titan Krios (FEI, Hillsboro, Oregon) with K2 direct electron detecting camera (Gatan, Pleasanton, California) in video mode. The SerialEM package was used for semi-automatic data collection. The total dose was ~66 electrons/Å$^2$, distributed over 40 video frames. Defocus was systematically varied from approximately 1 to 2.5 µm for the first 10 frames and then increased by 6 µm for the rest of the video. The data set was collected at ~81K magnification using the camera's super-resolution mode with an effective pixel size of 0.68 Å.

## Image processing: conventional helical analysis

Following drift analysis of the whole-image videos (*Li et al., 2013*), the first ten video frames and the remaining thirty were aligned and averaged separately. The resulted images are called low-defocus framesets and high-defocus framesets. Conventional helical analysis was performed with the low-defocus framesets.

Overlapping boxes covering each microtubule were manually selected from the low-defocus framesets. The CTFFIND3 algorithm was used to determine defocus and astigmatism parameters (*Mindell and Grigorieff, 2003*). Initial reference models of microtubules with 12–14 protofilaments were generated as previously described (*Shang et al., 2014*; *Sindelar and Downing, 2010*). Multi-reference analysis showed that 13-protofilament microtubules are the most numerous in the dataset. Based on this result, 13-protofilament microtubules were selected for further processing. After the initial 3D reconstruction, PDB model of kinesin-tubulin complex (PDB ID: 3J8Y) was fitted into the electron density map using Chimera (*Pettersen et al., 2004*) and atom coordinates relative to the microtubule axis were saved in a new PDB file. This new PDB model was then used to generate new initial reference models that are a closer match to the experimental microtubules and the multi-reference analysis was repeated.

Alignment of boxed particles was performed as previously described (*Shang et al., 2014*). Briefly, semi-exhaustive searching of helical parameters (XY-shifts and Euler angles) was performed with SPIDER (*Frank et al., 1996*). Seam location and polarity of every microtubule was determined by reference alignment as described (*Sindelar and Downing, 2007*). Following the initial refinement of helical parameters, the position of each 8 nm repeat of the microtubule was determined and a new stack of boxes was extracted and subject to another round of refinement. Thus, the final dataset contained one box for every 8 nm repeat of the selected microtubules.

XY-shifts and Euler angles of boxes along one microtubule are not supposed to change abruptly and boxes that are outliers may result from failed SPIDER refinement. To prevent the undue effects of outliers on further analysis, XY-shifts and Euler angles along each microtubule were subject to smoothing with least trimmed squares (LTS), which is a robust regression method (*Rousseeuw, 1984*). In practice, the XY-shifts and Euler angles are considered as individual functions of the numbering of boxes along one microtubule. The window size used for LTS includes five consecutive boxes. Regression is performed with all combinations of three out of the five boxes. The XY-shifts and Euler angles of the middle box are then replaced with the expected values calculated from the best regression. After LTS smoothing, if any two neighboring boxes have one of the Euler angles differ by more than five degrees, we mark a discontinuity point between them. Microtubules are broken into segments at the discontinuity points and segments shorter than 10 boxes are discarded. Boxes from the remaining microtubule segments entered the next stage: 3D reconstruction and refinement with FREALIGN (*Alushin et al., 2010*; *Grigorieff, 2007*).

FREALIGN refinements were performed as previously described (*Shang et al., 2014*), with the following modifications. A total of 13 rounds of refinement and reconstruction were performed, using successively higher resolution cutoffs for the refinement target (10, 8, 6, 5 Å). FSC calculations following the final round of FREALIGN refinement indicated that the resolution was approximately 5 Å (0.143 criterion). A total of 70,142 8 nm repeats were used. After 13 rounds, no further improvement of resolution could be achieved.

## Image processing: FINDKIN

After the final round of FREALIGN refinement, the XY-shifts and Euler angles of all 8 nm repeats in the low-defocus framesets were saved. In order to map the same 8 nm repeats onto the high-defocus framesets, whole-image cross-correlation was used to measure the shift between the low-defocus and high-defocus framesets and the latter were aligned with the former. A new stack of boxes, each corresponding to one 8 nm repeat, was then extracted from the high-defocus framesets. Because alignment by whole-image cross-correlation may not be sufficiently accurate, FREALIGN was then used to perform a local refinement of XY-shift for each box from high-defocus framesets, followed by LTS smoothing. The FREALIGN refinement-LTS smoothing was repeated for seven rounds.

For each box, the XY-coordinates and orientations of every tubulin dimer was calculated from the XY-shifts and Euler angles of the corresponding 8 nm repeat, according to a parametric model of 13-protofilament microtubule (*Chrétien and Wade, 1991*). For each potential kinesin-binding site (i.e. a tubulin dimer), we projected a PDB model of kinesin-tubulin complex or tubulin dimers alone (PDB ID: 3J8Y) to generate reference images of just this site with or without kinesin. The reference images were low-pass filtered to 20 Å.

To determine the search area for the tubulin dimer under examination, we compared the reference image of this site (with kinesin) against reference images of all other sites (with kinesin) in the same box with cross-correlation. The search area is composed of all the sites that give a cross-correlation score above a designated cutoff of 0.5 that limits the number of overlapping sites to a maximum of around 10 in our system.

Next, we generated reference models corresponding to the enumeration of all possible kinesin-binding patterns in the search area by combinatorially adding reference images of the involved sites. For each site, there are two reference images (with or without kinesin), and thus for a search area of N sites, there are a total of $2^N$ reference models.

Finally, cross-correlation scores were computed between the experimental image and reference models of all possible binding patterns. The closest matched pattern is selected as the one that gives the highest score. A binding state (occupied or empty) is then assigned to the tubulin dimer under examination in accordance with its counterpart in the best-matched pattern. This process was repeated for all the tubulin dimers and their collective binding states were recorded in kinesin-binding matrices, where '1' stands for a tubulin dimer bound with kinesin and '0' stands for the opposite.

The above binding pattern matching was performed on boxes from both low-defocus framesets and high-defocus framesets.

## Image processing: FINDKIN post-processing

From the binding matrices generated from high-defocus framesets, we selected all patterns of '0110' along a single protofilament. Next, we check the corresponding sites from the low-defocus binding matrices. If either one of the two flanking empty sites is occupied ('1???' or '???1' in the low-defocus binding matrices), this site is discarded. This procedure strongly favors the selection of isolated kinesin dimers, because the flanking empty sites are corroborated with additional information from the low-defocus framesets. Note that only the flanking sites and not the two sites in the middle are examined from the low-defocus binding matrices, and thus this procedure will not introduce model bias to the final reconstruction. These sites give us the cleanest kinesin dimer reconstruction.

Finally, low-defocus boxes containing the selected '0110' sites were used for 3D reconstruction. The XY-shift and Euler Angles of these boxes were available as in the last round of FREALIGN refinement. For the reconstruction, a 13-fold pseudo-helical symmetry was applied to transform all selected '0110' sites onto a single protofilament, as previously described (*Sindelar and Downing, 2007*). A total of 17862 asymmetric units were used. No further FREALIGN refinement was performed. In addition to '0110' sites, we also selected '010' and '011110' sites from the binding matrices. 3D reconstruction was performed similarly to the '0110' reconstruction. A total of 44298 asymmetric units were used for the '010' reconstruction. A total of 4831 asymmetric units were used for the '011110' reconstruction

The above image processing procedures are summarized in flowcharts shown in *Figure 1—figure supplement 3*.

## FSC calculations

In order to assess the resolution of kinesin in our maps (*Figure 3—figure supplement 1*), the FSC was calculated within a soft edge mask around the kinesin density. The kinesin chain of a PDB model (3J8Y) was fit into the kinesin density and a volume (low-pass filtered to 20 Å) was generated from the PDB. The density threshold was set so that the volume size is 25750. This volume was used as the mask and was softened with an edge width of 3 pixels.

The overall quality of the observed secondary structural features seen in our kinesin dimer maps seems to be substantially worse than would be expected, given the resolutions reported by our FSC calculations (~7 Å for '010' and '0110' reconstructions, ~9 Å for the '011110' reconstruction). It is therefore probable that our resolution estimates are affected by some degree of over-refinement in the maps, which could be addressed in future versions of FINDKIN by decreasing the sensitivity of FSC calculations to over-refinement and/or utilizing improved 3D refinement methods that reduce the level of over-refinement.

## Generation of synthetic microtubules

A parametric model of a 13-protofilament microtubule (*Chrétien and Wade, 1991*) was applied to PDB models of kinesin-microtubule complex and tubulin dimers alone (PDB ID: 3J8Y) to generate microtubule models with a mixture of occupied and empty sites. The occupied sites were randomly selected but were always in pairs to mimic kinesin dimers. Total numbers of occupied sites were controlled for different levels of occupancy (10, 20, 30, 40%). Five images of synthetic microtubules were generated for each occupancy level. Next, a layer of white noise was added to the images to mimic the noise from solvent background. Five different noise levels (1 through 5 with solvent signal to noise ratios of 4, 2, 1, 0.5 and 0.25 respectively on and around the area of microtubule) were applied to each image. Then a contrast transfer function (CTF) was applied to each image with a defocus randomly selected between 1.0 and 2.5 µm. For each image, we increased the defocus by 6 µm to generate a high-defocus counterpart. Finally, another layer of white noise was added to the CTF-adjusted images to mimic other sources of image noise, like the electron shot noise (noise level 1 through 5 for image signal to noise ratios of 0.4, 0.2, 0.1, 0.05 and 0.025 on and around the area of microtubule).

We generated 200 synthetic images (4 occupancy levels $\times$ 5 images per occupancy $\times$5 noise levels $\times$ 2 defocus) in total. They were subject to image processing similar to experimental images.

## Molecular graphics, rigid-body fitting and cross-correlation calculations

All figures were rendered with Chimera. Rigid-body fitting of domain structures into EM density maps and calculations of cross-correlation between EM maps are performed with Chimera. For the subdomain fitting, density maps were created from the PDB models at 9 Å resolution, and refinement of the coordinates was done by maximizing the cross-correlation with the head density. Subdomain rotations are measured with the 'measure rotation' function in Chimera (*Pettersen et al., 2004*).

To measure the cross-correlation between our dimer maps and published maps of monomeric kinesins, all maps were first aligned using the tubulin densities. Then the tubulin chains of a PDB model (3J8Y) were fit into the dimer maps and its kinesin chain was used to generate a volume low-pass filtered to 20 Å (with the 'molmap' function in Chimera). This volume was rendered at the threshold level of 0.1 and used as a mask to include only kinesin densities from the maps (with the 'mask' function in Chimera). Cross-correlation scores were computed between our dimer maps and the monomeric kinesins for the regions within the mask, including kinesin densities while excluding tubulins. Because the maps may have different background levels or are not on identical scales, the 'correlation about mean' function was used to compute the scores in Chimera (*Pettersen et al., 2004*).

## Accession codes

Cryo-EM maps for dimeric kinesin in a two-head-bound state (EMDB ID: 8546) as well as a singly-bound state (EMDB ID: 8547) on microtubules have been deposited (URL: http://emsearch.rutgers.edu/atlas/8546_summary.html and http://emsearch.rutgers.edu/atlas/8547_summary.html, respectively).

## Acknowledgements

We gratefully acknowledge Kirsten Knecht for her contributions in the purification of K420 protein and Sarah Rice for the generous gift of the K420 construct. We thank staff in the Yale CryoEM facility and High-Performance Computing facility for their maintenance of these facilities; and the Janelia Farm cryo-EM facility for data collection time on the Krios 1 microscope.

## Additional information

### Funding

| Funder | Grant reference number | Author |
|--------|------------------------|--------|
| National Institutes of Health | R01 GM 110530-01 | Charles V Sindelar<br>Daifei Liu<br>Xueqi Liu<br>Zhiguo Shang |
| American Cancer Society | ACS-IRG-58-012-55 | Charles V Sindelar<br>Zhiguo Shang |

The funders had no role in study design, data collection and interpretation, or the decision to submit the work for publication.

### Author contributions

DL, Data curation, Software, Formal analysis, Validation, Investigation, Visualization, Methodology, Writing—original draft, Writing—review and editing; XL, Software, Formal analysis, Methodology; ZS, Software, Investigation, Methodology; CVS, Conceptualization, Software, Formal analysis, Supervision, Funding acquisition, Investigation, Visualization, Methodology, Project administration, Writing—review and editing

### Author ORCIDs

Charles V Sindelar, http://orcid.org/0000-0002-6646-7776

## Additional files

### Major datasets

The following datasets were generated:

| Author(s) | Year | Dataset title | Dataset URL | Database, license, and accessibility information |
|-----------|------|---------------|-------------|--------------------------------------------------|
| Liu D, Liu X, Shang Z, Sindelar CV | 2017 | Dimeric Kinesin-1 on Microtubules with ADP-AlFx | http://emsearch.rutgers.edu/atlas/8546_summary.html | Publicly available at the Electron Microscopy Data Bank (accession no: EMD-8546) |
| Liu D, Liu X, Shang Z, Sindelar CV | 2017 | Kinesin-1 Binding Microtubules with ADP-AlFx in a One-head-bound State (low-pass filtered to 8.5 angstrom) | http://emsearch.rutgers.edu/atlas/8547_summary.html | Publicly available at the Electron Microscopy Data Bank (accession no: EMD-8547) |

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
