## [Decision Letter]

Thank you for submitting your article "Structural basis of cooperativity in kinesin revealed by 3D reconstruction of a two-head-bound state on microtubules" for consideration by *eLife*. Your article has been reviewed by two peer reviewers, and the evaluation has been overseen by a Reviewing Editor and John Kuriyan as the Senior Editor. The following individual involved in review of your submission has agreed to reveal her identity: Eva Nogales (Reviewer #3).

The reviewers have discussed the reviews with one another and the Reviewing Editor has drafted this decision to help you prepare a revised submission.

This manuscript addresses a long-standing and fundamental question regarding the structure and mechanism of the kinesin motor. This paper is a technical tour de force, highly innovative, in which the authors have developed a completely new methodology to be able to address a key question for our understanding of cytoskeleon motors: what are the structural states of a dimeric motors as it interacts with their track. All previous studies dealt with just monomer constructs, or with dimers under the usual saturation conditions that are used in microtubule (or actin) cryo-EM studies of bound proteins. Such conditions allow the use of the filament symmetry to be imposed for obtaining the motor structure, but naturally lead to the second head not being able to interact with the track due to full occupancy by the first one. Working under substoichiometric conditions that allowed for second head binding, required the development of new image processing strategies. Sindelar and colleagues have done just that.

We have a few points that would help to clarify the manuscript:

The second paragraph of the subsection “Leading and trailing heads exhibit distinct chemical and conformational states” reports the authors directly resolve weak density corresponding to the docked conformation of the neck linker. However, the corresponding figure (4B) does not clearly show this. The difference in density between the two maps could be shown more clearly (perhaps by zooming in and using a mesh representation of the EM map). If the figure is to remain unchanged, the wording in the text should be revised.

Figure 3 – The neck density appears to be much closer to the trailing head than the leading head. Can this be modeled with regards to the length of the two neck linkers? i.e. can the leading head neck linker reach back this far? Can the authors speculate why the neck would not adopt a position equidistant to the two motor domains?

I am somewhat unconvinced by the cross-correlation data (subsection “Leading and trailing heads exhibit distinct chemical and conformational states”, last paragraph), namely with the differences being so small. Figures to illustrate docking of the synthetic volumes in both heads would help support this point. Furthermore, EM data filtered/sharpened based on the 0.143 threshold is still relatively noisy – perhaps the effect of simulated noise on the synthetic maps could also help prove the cross-correlation differences to be significant.

---

## [Author Response]

*We have a few points that would help to clarify the manuscript:*

*The second paragraph of the subsection “Leading and trailing heads exhibit distinct chemical and conformational states” reports the authors directly resolve weak density corresponding to the docked conformation of the neck linker. However, the corresponding figure (4B) does not clearly show this. The difference in density between the two maps could be shown more clearly (perhaps by zooming in and using a mesh representation of the EM map). If the figure is to remain unchanged, the wording in the text should be revised.*

To address this concern, we added an additional panel (Figure 4, legend) that shows a close-up view of the neck linker. We fit a crystal structure of monomeric kinesin in the ATP-like state with a docked neck linker (PDB ID: 4HNA) into our leading and trailing head densities and the neck linker from the PDB is shown as a yellow strand. In the trailing head, portions of the head density overlap with the yellow strand, but not in the leading head.

We did the same analysis on the parallel reconstruction of kinesin dimer derived from the ‘011110’ pattern, and the result is consistent. This is shown in Figure 4—figure supplement 1.

*Figure 3 – The neck density appears to be much closer to the trailing head than the leading head. Can this be modeled with regards to the length of the two neck linkers? i.e. can the leading head neck linker reach back this far? Can the authors speculate why the neck would not adopt a position equidistant to the two motor domains?*

We agree that this is an important point; we offer a simple explanation for this observation. The neck linker originates at the C-terminal end of helix α6, as indicated in Figure 3. The coiled coil density is about equidistant from the ends of helices α6 in both leading and trailing heads and the neck linker can indeed extend from the coiled coil to the leading head, as shown in Figure 3. Because the end of helix α6 is closer to the minus-end side of the motor domain, the coiled coil density thus appears closer to the trailing head. We therefore believe there is a no need to invoke a more complicated explanation for the position of the neck coiled coil. We addressed this point in the sixth paragraph of the subsection “Application of FINDKIN to experimental dataset reveals dimeric kinesin on microtubules” and also in the figure legend for Figure 3.

*I am somewhat unconvinced by the cross-correlation data (subsection “Leading and trailing heads exhibit distinct chemical and conformational states”, last paragraph), namely with the differences being so small. Figures to illustrate docking of the synthetic volumes in both heads would help support this point. Furthermore, EM data filtered/sharpened based on the 0.143 threshold is still relatively noisy – perhaps the effect of simulated noise on the synthetic maps could also help prove the cross-correlation differences to be significant.*

We thank the reviewers for highlighting this clear point of concern in the original submission. We have now conducted a much more careful cross-correlation analysis; among the improvements in our methodology were: (1) utilizing the original cryo-EM maps for the monomer kinesin structures, rather than PDB models, in order to compare with our dimer maps; (Andreasson et al., 2015) extending the cross-correlation analysis to both of our independent kinesin dimer maps, in order to assess the significance of the scores; and (3) comparing the leading and trailing heads of the dimers with each other, to assess how significant the structural differences between these were. In the process, we discovered that the monomer cryo-EM maps discriminate between our dimer head densities quite a bit more effectively than the corresponding PDB models – most likely because the PDB models fail to capture a variety of phenomena (B-factors/breathing movements, disordered regions, inaccuracies in the modeled electrostatic potential) that occur in the imaged EM sample.

The resulting calculations are much more definitive in demonstrating that the trailing head correlates more closely with monomer kinesin in the ATP-like state, while the leading head correlates more closely with monomer kinesin in the no-nucleotide state. Our comparison of the two independent dimer reconstructions reveals a tight clustering of like heads with each other (trailing/trailing, leading/leading, trailing/ATP, leading/apo). The results are illustrated in the new Figure 4—figure supplement 3. The new calculations are presented and discussed in subsection “Leading and trailing heads exhibit distinct chemical and conformational states”, first paragraph, subsection “Leading and trailing heads exhibit distinct chemical and conformational states”, last paragraph, subsection “Contamination with singly-bound heads may largely explain discrepancies between the kinesin dimer map and structures of monomeric kinesins”, third paragraph and the method for the calculations in subsection “Molecular graphics, rigid-body fitting and cross-correlation calculations”, last paragraph.